# Clinical Assessment of Chronic Musculoskeletal Pain—A Framework Proposal Based on a Narrative Review of the Literature

**DOI:** 10.3390/diagnostics13010062

**Published:** 2022-12-26

**Authors:** Helen Cristina Nogueira Carrer, Gisele Garcia Zanca, Melina Nevoeiro Haik

**Affiliations:** 1Department of Physical Therapy, Federal University of São Carlos (UFSCar), São Carlos 13565-905, Brazil; 2Interdisciplinary Center for Pain Care, Federal University of São Carlos (UFSCar), São Carlos 13565-905, Brazil; 3Department of Physical Therapy and Occupational Therapy, São Paulo State University (UNESP), Marília 17525-900, Brazil

**Keywords:** pain assessment, musculoskeletal pain, pain management, international classification of functioning, disability, health

## Abstract

The assessment of chronic musculoskeletal pain (CMP) is a challenge shared by several health professionals. Fragmented or incomplete assessment can cause deleterious consequences for the patient’s function. The objective of this paper was to propose a framework for clinical assessment of CMP based on the current literature and following the conceptual model of the International Classification of Functioning and Health (ICF). We propose that the ICF rationale may help to guide the processes, acting as a moderator of the clinical assessment, since it changes the perspective used to obtain and interpret findings during anamnesis and physical examination. Additionally, updated specific knowledge about pain, including that of pain domains and mechanisms, along with effective patient–clinician communication may act as a mediator of CMP assessment. We conduct the readers through the steps of the clinical assessment of CMP using both the proposed moderator and mediators and present a clinical example of application. We suggest that the proposed framework may help clinicians to implement a CMP assessment based on the biopsychosocial model using a critical and updated rationale, potentially improving assessment outcomes, i.e., clinical diagnosis.

## 1. Introduction

Chronic musculoskeletal pain (CMP) is a common public health problem that causes a significant impact on patient health, quality of life, and functioning. It is one of the leading causes of years lived with disability in the world [1]. For example, low back pain, headaches, and neck pain were among the 10 leading causes in 2017 for both sexes [1,2]. Assessing individuals with chronic musculoskeletal pain can be challenging for different health professionals working in pain management. To this end, clinicians must organize a vast amount of individualized and personal information, making the assessment of each individual a unique experience [3]. Furthermore, based on a sound evaluation process—that is, a continuous action—the exchange of information and experiences between patient and professionals will be important for screening purposes and clinical diagnosis. In other words, establishing an appropriate evaluation process for pain management requires time and mutual willingness to consolidate a rapport and therapeutic alliance [4].

Considering that CMP is a multidimensional and complex clinical condition, current recommendations for CMP assessment are based on the biopsychosocial model of healthcare, in which clinicians must adopt a practical integrated management approach that should begin with an assessment focused not only on pain but on the whole person [5]. Clinical guidelines recommend assessing disability as the highest priority when dealing with individuals with chronic pain [6]. It is important to note that individuals with CMP and higher disability are eight times more likely to seek care than those with low disability [7]. Biological aspects—such as range of motion and muscle strength—and psychological factors—such as self-efficacy, catastrophizing and kinesiophobia—have been shown to be potentially related with the levels of pain and disability in CMP [8,9]. Moreover, social factors, such as participation and self-perceived ability to participate in usual roles, are predictors of pain interference in individuals with CMP [10]. Sociodemographic aspects, such as race, ethnicity and culture, have also been related to disability through coping strategies, pain beliefs, or self-efficacy levels [11].

The biopsychosocial model may be implemented considering the International Classification of Functioning, Disability, and Health (ICF) conceptual model by addressing the disability overview of the person seeking for care [12]. The use of the ICF conceptual model has been widely disseminated in clinical settings as a classification of disability and an outcome evaluation instrument [12,13]. Accordingly, clinicians should manage information related to the health condition and individual functioning and quality of life to build a comprehensive clinical assessment [5]. Despite the clear need of integrated care considering the whole person, a comprehensive assessment may not be so clear and easy to perform [14]. In this paper, we suggest that the ICF conceptual model may act as an important moderator of the CMP assessment, allowing for a broader perspective to achieve clinical diagnosis [15,16].

An approach for evidence-based diagnosis has been made practical by the public-private partnership of Analgesic, Anesthetic, and Addiction Clinical Trial Translations, Innovations, Opportunities, and Networks (ACTTION) with the US Food and Drug Administration and the American Pain Society (APS) [17]. This effort, called the ACTTION-APS Pain Taxonomy (AAPT), suggests a heuristic model for conducting pain assessment. It encompasses the understanding of the pain-related characteristics and their underlying mechanisms based on the persons’ perception of their own health condition [18,19], which is considered a gold standard for pain assessment [7]. Therefore, we further suggest that assessing specific pain domains and pain-specific self-reports using empathic communication may act as a mediator of CMP assessment [18].

Considering the above-mentioned aspects, the aim of this paper was to propose a framework for clinical assessment of CMP based on the current literature and outlined following the conceptual model of the International Classification of Functioning and Health (ICF), as illustrated in Figure 1. The framework’s purpose is to guide clinical reasoning used in the assessment process in order to identify and organize the most significant demands of persons with CMP, improving clinical diagnosis.

## 2. Methods

In order to identify, analyze, and include assessment approaches and tools suited to the ICF conceptual model, we performed a narrative review and synthesis of the peer-reviewed literature on CMP. A narrative review interprets the contents of several research papers using a thorough, critical, and objective analysis to synthesize and describe previously published data [20,21]. The six components necessary for a quality review were examined for this study according to the Scale for the Assessment of Narrative Review Articles (SANRA). To review the literature, we adopted a staged process. The initial step was to find original research texts that (a) focused on CMP and (b) included ICF-based assessment. To establish the search strategy, initial keywords were generated for each conceptual category of the study aim. Keywords from three categories (pain assessment OR musculoskeletal assessment OR functioning assessment OR pain management; chronic musculoskeletal pain; international classification of functioning, disability, and health) were identified and used to build search strings in the PubMed database. Searches were limited to English-language publications and to a publication period (January 2002–December 2022), as selected articles had to be published after the ICF was released in 2001. We aimed to highlight issues closely related to clinical practice and, therefore, no rigorous inclusion or exclusion criteria were applied. Accordingly, we also considered secondary research (such as systematic reviews and scoping reviews) as well as editorials and commentaries. Main concepts that emerged from the literature were organized, and the relationships between them were used to build a framework proposal to help clinical implementation of a CMP assessment following this rationale. 

## 3. The Framework Proposal

### 3.1. The ICF as a Moderator of CMP Assessment 

Moderators are elements or characteristics that affect the study’s subject and are situated between the independent and dependent variables in the statistical field [22]. The ICF reasoning helps the clinician to guide, systematize, and standardize the countless pieces of information identified during anamnesis and physical examination, which encompasses individuals’ ability to interact in their social life environments (familiar, professional, and recreational) physically and psychologically [15]. Figure 2 illustrates how ICF reasoning may modify CMP assessment and change clinical reasoning. 

In individuals with CMP, scientific evidence validates the assessment of individuals’ functioning as a priority over pain intensity or deficiencies in body structures and functions [23]. For example, among people with lower back pain, 40% of the complaints involve limitations in the performance of activities and restrictions in social participation, such as changing or maintaining body position, recreation and leisure, lifting and carrying objects, or paid employment. On the other hand, only 26% of complaints referred to deficiencies in body functions and 2% in body structures. Furthermore, in this population, 17% of complaints are still influenced by environmental factors and 11% by personal factors [24]. In people with musculoskeletal shoulder pain, the scenario is similar: 87% of the complaints involve categories of activity and social participation—such as recreation and leisure, lifting and carrying objects, performing household chores, paid employment, and using the hand and arm—while only 10% of complaints refer to impairments related to body structures and functions, and 2% involve environmental factors [25].

Therefore, when applied to CMP assessment, the ICF conceptual model may change the perspective used to obtain and interpret findings during anamnesis and physical examination because of a global “cultural change” [26,27]. This change of perspective has the potential to change CMP assessment outcome, i.e., the clinical diagnosis. The World Health Organization Disability Assessment Schedule (WHODAS) is an example of a useful tool for assessing a variety of performance-based outcomes of functional status. This might provide an alternative and more general assessment of CMP [28] since it was developed using a comprehensive set of items from the ICF considering six domains (cognition, mobility, self-care, getting along, life activities, and participation) containing 36 items. A first step could be the use of the 12-item WHODAS, which provides an individual functional profile and is an adequate, internally consistent, and reliable multidimensional measure highly correlated with other measures of disability [29]. A recent scope review shows that those with chronic low back pain report limitations on self-care, life activities, and getting along with people measured with WHODAS [29], which corroborates with previous reports of functioning complaints [24]. Even though there is not a large amount of evidence about functioning scores measured with WHODAS in people with CMP, clinicians may be able to understand clearly which part of the disability should be evaluated more thoroughly when using the instrument to map the level of functioning, as suggested by [30] for the physical aspect of functioning.

### 3.2. Mediators of CMP Assessment: Specific Assessment of Pain and Effective Communication

A mediator is a go-between for two variables. For example, mobility (an independent variable) can affect work achievement (a dependent variable) through the mediation of pain intensity [31]. A mediation relationship may offer a more comprehensive understanding of the relationship between the two variables. In the present proposal framework, we consider that specific assessment of pain may guide anamnesis and physical examination to improve clinical diagnosis of CMP, acting as a mediator. The specific assessment of pain should contain CMP domains, such as intensity or severity, onset and location of the symptoms, duration of the symptoms, provocative and relief factors, predominant pain mechanism, and other associated problems. 

Furthermore, the use of patient-reported outcome measures (PROMs) may be an important strategy built into the mediation process of the CMP assessment [32]. PROMs may help to collect important information for understanding such a complex and multidimensional condition as CMP. These tools value persons’ perceptions about their own condition but provide these data using standardized measures that can also be used for following up intervention effects, i.e., WHODAS. Nevertheless, standardized tools do not overcome the importance of communication abilities. Effective communication, which involves verbal and nonverbal forms of expression, should help persons to identify relevant aspects in their lives and, at the same time, create spaces for reflection on the veracity of some dysfunctional behaviors. It may be challenging to transition between the archetypes of the patient–health professional connection due to extra obstacles. For example, a poor relationship might result between a patient immersed in a social culture that requires people suffering from adopting a passive role and a professional workforce with undeveloped empathy [33]. Besides the relationship between patient and health professional, other aspects of communication deserve attention in cases of chronic pain, to strengthen therapeutic alliance and prevent nocebo effects [34]. Figure 3 summarizes the mediation role of specific assessment of pain (including pain domains and PROMs), and effective communication on CMP assessment. The following sessions present suggestions to include them during anamnesis and physical examination. 

## 4. Current Concepts on Anamnesis for CMP Assessment

Anamnesis is a standard procedure in health services and, when properly conducted, is responsible for up to 85% of the clinical diagnosis. In comparison, clinical examination findings diagnose about 10% of cases and complementary exams only 5% [35]. Before characterizing the pain, itself, the analysis of the current history is also very important in making other decisions, such as referral in cases suggestive of red flags—that is, warning signs identified in the anamnesis that raise the suspicion of potentially serious conditions. For the specific assessment of CMP, anamnesis should include the evaluation of pain domains. Also, moderated by the ICF reasoning, the anamnesis should guide the clinician in identifying the relevant PROMs to be used. 

### 4.1. Pain Intensity or Severity

The Numerical Scale of Pain Intensity Assessment is one of the most suitable tools for recording pain intensity among adults [36], and pain intensity seems to be an important predictor of disability for several domains of life, measured by the 12-item WHODAS, particularly for physical domains of functioning [29,37]. However, considering only pain intensity constitutes a reductionist perspective about the impact that CMP may cause on a person’s life. The focus on the sensitive and discriminatory aspects of pain may undervalue changes following interventions. From this perspective, tools that assess the multidimensional aspects of pain can provide new insights into critical clinical outcomes for the patient. The Brief Pain Inventory (BPI) rapidly assesses the severity of pain and its impact on functioning. In addition, it is available in approximately twelve languages [38]. The McGill Pain Questionnaire (PMQ) assesses both quality and intensity among three domains of subjective pain: sensory intensity, cognitive evaluation of pain, and emotional impact of pain [39]. Finally, for neuropathic pain, pain DETECT screening questionnaire identifies the prevalence of pain components in pain conditions, and the Neuropathic Pain Symptom Inventory (NPSI) helps to discriminate and quantify five distinct clinically relevant dimensions of neuropathic pain: burning, pressing paroxysmal, evoked, and paresthesia/dysesthesia [40].

### 4.2. Onset and Location of Symptoms 

It should be determined whether pain started suddenly or insidiously. For example, a sudden onset of intense pain with no clear provocation may indicate the need for urgent medical attention depending on the body region, such as the abdomen. To better identify pain location, most indicated tools involve body maps on which individuals can paint the involved body regions. The BPI [41] and the Nordic Musculoskeletal Symptom Questionnaire [42] are examples of questionnaires that include body maps, but other standardized body sketches could be used for this purpose.

### 4.3. Duration and Chronicity 

Regardless of age, gender, or socioeconomic condition, many individuals have had one or more episodes of musculoskeletal pain at some point in their life [43]. Identifying pain duration is essential for the clinical classification of chronic pain. Questions about the onset of symptoms characterize the duration of the complaints. In cases of chronic pain, typically defined as when pain lasts for more than three months, health professionals should look for broader aspects of causality and not just those arising from physical disorders [44,45]. Clinicians should be aware to identify possible association of pain history and other significant events of persons’ lives. Active listening is important during all anamnesis procedures but may be especially remarkable during this step. 

### 4.4. Provocative and Relieving Factors 

Investigating the provocative and relieving factors of pain symptoms is very important for the clinical diagnosis of a health condition and for identifying the profile, beliefs, and behaviors adopted by the person [46]. For example, consider a person that complains about lower back pain radiating to the gluteus and lower limb and reports pain worsening when standing and relief when lying down for less than 10 minutes. While the reported pain pattern suggests sciatica, it may also reveal some intrinsic beliefs of the patient, such as that the rest is good and that the effort is deleterious to the condition. When assessing these factors, proper communication plays a key role since it can encourage or hinder the person’s involvement [47,48,49]. A recent study of techniques for improving clinician–patient communication on chronic pain showed that patients’ main goal is to be heard [50]. When engaged in partnership building and supportive talk, clinicians create opportunities for persons to discuss their needs and to be involved. 

### 4.5. Predominant Pain Mechanism 

Three predominant mechanisms are currently recognized: (1) nociceptive, which occurs due to the activation of peripheral nociceptors from non-neural tissue; (2) neuropathic, which arises from lesions or diseases affecting the somatosensory nervous system; and (3) nociplastic, which is related to altered nociception mechanisms with no clear evidence of the other two mechanisms. The predominant mechanism may be suggested by analyzing the individual’s history. There are standardized tools to help professionals to identify some aspects, especially regarding neuropathic pain. The DouleurNeuropathique 4 (DN4) is a screening tool that includes questions related to symptoms and aspects of physical examination and may help to identify neuropathic pain [49]. There are also PROMs that may be useful for this purpose, such as those previously mentioned: painDETECT [40] and the McGill Pain Questionnaire [39]. Aspects identified during anamnesis should be interpreted associated with others assessed during physical examination, as described later. 

### 4.6. Other Associated Problems 

It is critical to consider the daily routines of CMP patients since they may impact on pain-related aspects and their response to treatment. Sleeping habits should be investigated since inadequate patterns of sleep may, among other effects, increase pain sensitivity [50]. On the other hand, the presence of CMP can also be responsible for sleep fragmentation, which is related to fatigue and more pain [51]. The Pittsburgh Sleep Quality Index is one of the most-used PROMs to detect alterations on sleep patterns, helping clinicians to advise their patients about strategies to improve it as an important part of pain treatment [52]. All clinicians involved in CMP care should be qualified to deliver general sleeping hygiene orientation for patients.

Sleeping alterations are commonly related to mental health impairment as well. The literature points to a frequent coexistence of symptoms of anxiety and depression among individuals suffering from chronic pain [53,54,55]. The Patient Health Questionnaire-PHQ 9 [56] and the Generalized Anxiety Questionnaire GAD-7 [57] are examples of short PROMs that may be used for screening anxiety and depression symptoms. Depending on their findings, clinicians should consider the relationship of these aspects with pain and, in cases that major disorders are suspected, refer to a mental healthcare professional. 

Furthermore, inadequate strategies for coping with pain may be related to symptom amplification, fear of movement, negative thoughts, low self-efficacy, constraints on social involvement, and negative future expectations [58]. Maladaptive beliefs and hypervigilant behavior are associated with a poor prognosis regardless of interventional technique [59]. Health professionals should be able to recognize signals of these strategies during anamnesis. In suspected cases of inadequate coping strategies, standardized tools may help to identify and follow the levels during the treatment course. Some examples of recommended PROMs are the fear-avoidance questionnaire, FABQ [60]; the Tampa Kinesiophobia Scale [61]; and the Negative and Positive Affect Scale, PANAS [62]. It is important to highlight that while the scores are important to follow, clinicians should also look at patients’ answers to specific questions to focus on the most critical aspects during rehabilitation. Figure 4 presents a model with questions that may help to identify other frequent psychosocial aspects among persons with CMP.

Identifying associated problems in individuals with pain also goes beyond the scope of individual aspects. For example, people with low socioeconomic status may face barriers that can contribute to perpetuating pain symptoms and restricting function levels [64,65].

After evaluating the current health condition of people with CMP, it is important to identify previous unsuccessful treatments [66]. This information is essential to find interventions that the patient will accept better since they may present a resistance to those that did not work well previously. Finally, screening for chronic noncommunicable diseases, such as diabetes, hypertension, dyslipidemia, dementia, and kidney disease, is vital to provide general guidelines for the control of the respective disease and for the risk management of future interventions to be prescribed for the treatment.

Some disorders, including osteoarthritis, ankylosing spondylitis, systemic lupus erythematosus, psoriatic arthritis, and even fibromyalgia, can be clinically diagnosed with the support of a family history of painful conditions [67]. However, health professionals should be aware of sedentary lifestyle, inadequate diet, and exposure to environmental factors reported by the patient, which may balance the causality of hereditary diseases.

## 5. Current Concepts on Physical Examination of CMP

While the role of psychosocial aspects on CMP has been increasingly recognized and should be comprehensively explored during anamnesis, the importance of physical examination should not be underestimated. People with CMP usually perceive that physical examination has a positive impact on the clinician–patient relationship [68]. Physical functioning consists of a person’s capacity in different situations considering the context and how barriers are managed or avoided [30]. In line with the proposal of the ICF perspective as a moderator of CMP assessment and with an integrated framework for clinical decision recently proposed [69], we suggest starting physical examination by assessing physical function. The identification of limited activities and restricted participation may guide the next steps of the assessment, encompassing observation, palpation, assessment of the range of motion, muscle performance, sensitivity, and special tests [70]. 

The mediators—specific assessment of pain and effective communication—may help the clinician to adjust physical examination to the person’s context, valuing the person’s perceptions, identifying the predominant mechanism of pain, enhancing the therapeutic alliance, and avoiding pain hypervigilance. Clinicians should be aware that the touch during examination, when performed gently and attentively, represents an expression of care and empathy and is also a form of communication that may contribute to supporting the patient [71]. Especially for individuals that present evidence of symptom amplification, clinicians should avoid questioning about pain at each procedure performed [72]. Instead, the clinician may ask the person to report any discomfort during the procedures, for example [72]. In addition, it is recommended not to cause unnecessary suffering during the examination to an individual with a condition of vulnerability and stress increased by the painful experience itself. Thus, clinicians should avoid procedures that are not useful to guide the clinical diagnosis and the treatment plan. 

### 5.1. Assessing Physical Functioning

Identifying limited activities and restricted participation is crucial to guiding physical examination since it will allow focusing on body structures and functions potentially involved. Moreover, the recent body of knowledge shows that the main goal of individuals with chronic pain is to improve function, which requires the physical function assessment as the mainstay to clinical decision making regarding the plan of care to improve activities and participation. We propose that the Patient-Specific Functional Scale (PSFS) is a good starting point for mapping physical functioning [73]. The PSFS is a PROM in which the person may list from three to five important tasks that have been impaired by pain, classifying the current ability to perform on a 0–10 scale, with lower scores representing greater difficulty. Therefore, the PFSF values personal functional goals [74]. Another possible strategy is using the Test Instrument for Profile of Physical Activity (TIPPA), which has been developed to map physical ability in individuals with CMP in a rehabilitation setting [75]. It has four parts and is recommended as part of a more overall and complex assessment [75].

Other PROMs may also help with the initial investigation of physical functioning impairments. Clinicians may utilize PROMs that are region-specific, such as the Disabilities of the Arm, Shoulder, and Hand (DASH); for a specific condition, such as the Western Ontario and McMaster Universities Osteoarthritis Index; or generic, such as the WHO Disability Assessment Schedule. However, it is important to consider that memory failure may impact the outcomes of these instruments [30] and that contextual factors may act as facilitators or barriers. 

Therefore, afterwards, it is also recommended to select appropriate physical performance tests to carry out in the clinical environment according to the limitations and/or restrictions identified through the PROMs [30]. Ideally, physical performance should be assessed in the environmental context in which the person usually performs it [69]. Negative beliefs may also influence self-perceived physical function and contribute to different findings between PROMs and objective performance tests [76]. 

### 5.2. Pain Mechanism

Identifying aspects that may indicate the predominance of one pain mechanism over the others may also help to guide physical examination. For example, the physical examination of a person with predominantly nociceptive pain should investigate deficiencies in body structures and functions. On the other hand, persons with predominantly nociplastic pain might report pain during different examination procedures involving several body regions [77]. Although there are still no proven valid and reliable methods to differentiate between pain mechanisms, experts recommend a set of indicative factors for each mechanism [78]. Findings such as pain reporting being directly and proportionally aggravated by specific movements, consistent pain reproduction patterns, and localized distribution of pain without generalized hypersensitivity indicate the predominance of a nociceptive mechanism. The neuropathic mechanism may be associated with pain distributed in a region compatible with peripheral nerve or dermatome anatomy, sensory deficits with a dermatome pattern, hypoesthesia, hypoalgesia, decrease or absence of deep tendon reflexes, positive Tinel tests (percussion in the nerve path), and muscle atrophy with a distribution compatible with myotome. The predominant nociplastic pain mechanism is usually related to regional rather than discrete pain distribution, a pain pattern that cannot entirely be explained by nociceptive or neuropathic mechanisms, and clinical signs of generalized hypersensitivity (such as mechanical, heat or cold allodynia, or painful after-sensations after any evoked pain hypersensitivity assessments). The mechanism may be classified as “probable nociplastic pain” when the person presents a history of pain hypersensitivity in the region of pain and at least one of comorbidity from a predefined set (increased sensitivity to sound, light and/or odors, sleep disturbance, fatigue, or cognitive problems) [79]. Findings that possibly exclude nociplastic pain are: a sensation of pain relief in movements that lead to structural decompression; absence of hyperalgesia in areas remote from the primary site of pain; presence of localized muscle atrophy; and efficient conditioned modulation of pain [78].

It is important to be aware that these factors are experts’ opinions and that there are still no valid criteria currently. However, they may work as hints for professionals, allowing them to focus on specific aspects that may help in clinical reasoning when combined with other findings of the assessment. 

### 5.3. Observation

Observation should involve more than just inspecting the painful body region. Clinicians should observe movement patterns when individuals are not aware they are under observation—for example, observe gait patterns when arriving at the clinic—since they may reveal important information. Frequent changes in position, facial expressions, and postures may indicate avoidance of specific joint movements, or apprehension to perform tasks, such as sitting, standing, or moving a particular segment [80]. These observations, carried out in an "informal" way during the presentation and anamnesis [69] may provide valuable clues regarding the structures possibly affected or avoided movements since patients may change movement patterns under formal examination.

### 5.4. Palpation

Palpation is traditionally used to detect changes in shape and texture of the structures, besides the mechanical stimulation that induces symptoms. It is also possible to identify changes in muscle tone during palpation, which indicates a mechanism of protection of the painful region. For some chronic rheumatologic diseases, temperature increase may indicate active inflammation. We recommend careful attention to nonverbal communication during palpation through gentle touching and avoiding comments about findings that may favor nocebo effects [71]. It is also fundamental to be aware of an individual’s verbal and nonverbal reactions during this step of examination [30].

### 5.5. Range of Motion (ROM) and Muscle Performance 

Joint mobility and muscle performance are frequently impaired in persons with CMP. The PSFS and the observation of functional tasks help to guide which joints and muscles are relevant to assess [30]. Restrictions during active ROM may point to muscle weakness or pain-induced inhibition. Muscle strength, endurance, or power should be considered during assessment depending on the required performance during functional activities reported as impaired. Muscle weakness may occur due to inhibition caused by pain, alteration in innervation, or disuse. In addition, muscle weakness with a plausible neuroanatomical distribution may indicate a neuropathic pain mechanism [78]. It may be important to assess pain intensity during individual contractions of the relevant muscles to follow evolution of the complaint during reassessments because the treatment strategy may improve body functions, such as muscle performance, despite pain intensity maintenance. 

### 5.6. Sensitivity Assessment

This step aims to identify alterations in sensitivity, allodynia, or hyperalgesia, important mediators of CMP pain assessment. Semmes–Weinstein monofilaments are used in the affected area in order to investigate a possible loss of or reduction in pressure sensitivity [81]. Questions about discomfort or pain perception while wearing clothing or accessories are typically used to evaluate allodynia [82]. Dynamic mechanical allodynia is a painful sensation caused by gentle movement using a cotton pad or soft brush. Static mechanical allodynia can be investigated with digital palpation using a pressure of approximately 4 kg (enough to cause examiner’s nail bed blanching). It can be considered allodynia when the individual reports pain under this pressure. Cold allodynia can be investigated by holding a metal object at a temperature of approximately 20 °C against the skin. For heat allodynia, the same object may be heated in water to a temperature of approximately 40 ºC and then held against the individual’s skin [82]. As mentioned before, these procedures may help with the identification of the predominant pain mechanism.

### 5.7. Specific Clinical Tests 

Neurodynamic tests may be helpful in cases of peripheral neuropathic pain aggravated in an area compatible with the innervation [78]. An example is the straight leg raise (SLR) test for the sciatic nerve (nerve roots L4 to S3). Investigation of peripheral neuropathic pain may also involve Tinel tests, which involve percussion over the area of the peripheral nerves. When neuropathic pain is suspected, evaluation of deep tendon reflexes is also suggested [78].

Several specific clinical tests have been described in the literature to assess the involvement of joint structures in CMP. Most of these tests were developed considering the pathoanatomical perspective and therefore aim to identify specific structures involved in pain by reproducing symptoms during specific movements. However, the validity and reliability of these tests are highly variable for many conditions. In addition, many tests present risks of detecting positive findings that do not truly represent the involved body structure [83,84]. Thus, the outcomes of these tests should be interpreted with caution and combined with other findings from the anamnesis and physical assessment.

## 6. An Example of Clinical Application of the Framework Proposal

The proposed framework for assessing the CMP requires professional skills based on the ICF classification as well as on current pain knowledge. Clinicians must consider each person’s functional and social demands allied with the must current evidence in pain neuroscience. The final product of the assessment should provide a clinical diagnostic as a singular result. Figure 5 summarizes clinical findings from the assessment of a middle-aged woman with chronic low back pain. It is important to highlight the significance of the ICF codes in establishing a universal language of clinical diagnosis. However, the current study does not aim to go into detail about how ICF is used. The case demonstrates how the classification could be combined with pain knowledge to highlight biopsychosocial elements in a CMP condition.

In the example provided, since the first interaction, the professional encouraged the woman to describe her history, recalling the onset of the pain and identifying both relieving and worsening events. The professional listened to the patient with attention, making eye contact and avoiding expressions of concern that could induce nocebo effects during all procedures [28,34]. The patient’s history revealed that pain may be related to changes in work activities. The BPI body map reveals that besides low back region, pain also extended to the right buttock. The BPI also revealed an intense pain that significantly affected the patient’s life activities. The most impaired domain of WHODAS 2.0 was mobility, with a score of 90%. This instrument is critical to the ICF’s proposal as a pain assessment moderator [28]. The DN4 indicated neuropathic mechanism, which was reinforced by findings of physical examination, such as the Positive Straight Leg Raise Test. Provocative factors included both work and leisure activities involving similar movements associated with handling and trunk flexion, which also produced pain during physical examination. Social factors act as facilitators since they make activities and participation possible. There was familiar support for housework and understanding when the patient needed resting for momentary pain relief. Furthermore, her work environment offered educational opportunities. Using the PSFS tool as a guide, the physical examination identified the most crucial activities that required further investigation, which was consistent with the findings from the WHODAS. Pain relief was detected during trunk extension, while the range of motion was painfully constrained during trunk flexion. Observation demonstrated limping after provocative testing and during significant exertion (walking). A lumbar muscular spasm was detected when it was palpated. It is important to emphasize how extensive the proposed framework for assessing the CMP works. We believe that applying this rationale for CMP assessment could contribute to planning individualized rehabilitation aiming at improving a person’s function level. 

## 7. Final Considerations

We proposed a framework for guiding clinical assessment of CMP based on the current literature of CMP and considering the ICF conceptual model, specific assessment of pain and effective clinician–patient communication. The ICF conceptual model acts as a moderator of CMP assessment, allowing clinicians to put into practice the biopsychosocial model. The perspective provided by the ICF may change the way that clinicians organize and interpret assessment findings and therefore may change outcomes, i.e., clinical diagnosis. 

However, CMP is a complex and multidimensional condition, requiring specialized knowledge to use specific tools, allowing the investigation of all pain domains and including patient-reported outcomes. Furthermore, therapeutic alliance, very significant for CMP care, depends on effective communication, which must be constructed during all clinician–patient interactions. Therefore, specific assessment of pain (using pain domains and PROMs) and effective communication act as mediators of CMP assessment since they allow for a more comprehensive understanding of the patient’s status. 

Strategies to improve the implementation of moderator and mediators of CMP assessment should include continuous education for health professionals [85,86]. Clinicians should be trained to assess and comprehend the biopsychosocial components of pain to assist individuals in managing their condition as effectively and as early as possible [87]. The framework proposed in this paper may assist clinicians in understanding and organizing current concepts to implement CPM assessment, improving its outcome, i.e., the clinical diagnosis.

## Figures and Tables

**Figure 1 diagnostics-13-00062-f001:**
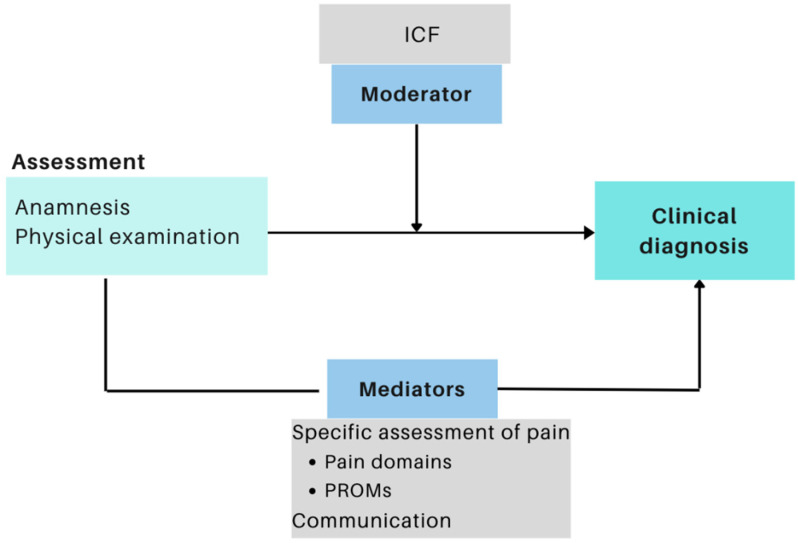
Proposal for a conceptual framework for assessing chronic musculoskeletal pain.

**Figure 2 diagnostics-13-00062-f002:**
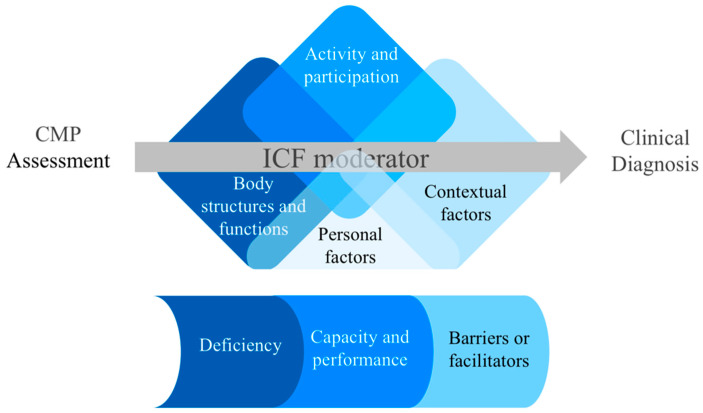
The ICF model as a moderator for clinical assessment of chronic musculoskeletal pain (CMP), able to modify clinical diagnosis.

**Figure 3 diagnostics-13-00062-f003:**
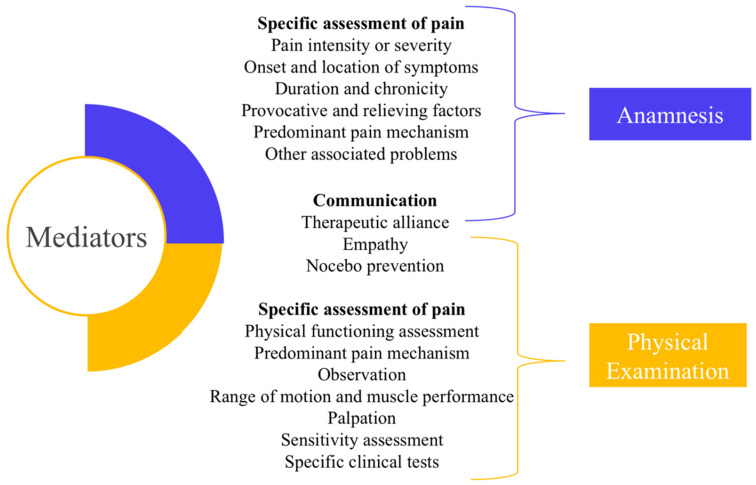
Specific assessment of pain, and communication act as mediators during anamnesis and physical examination for chronic musculoskeletal pain assessment.

**Figure 4 diagnostics-13-00062-f004:**
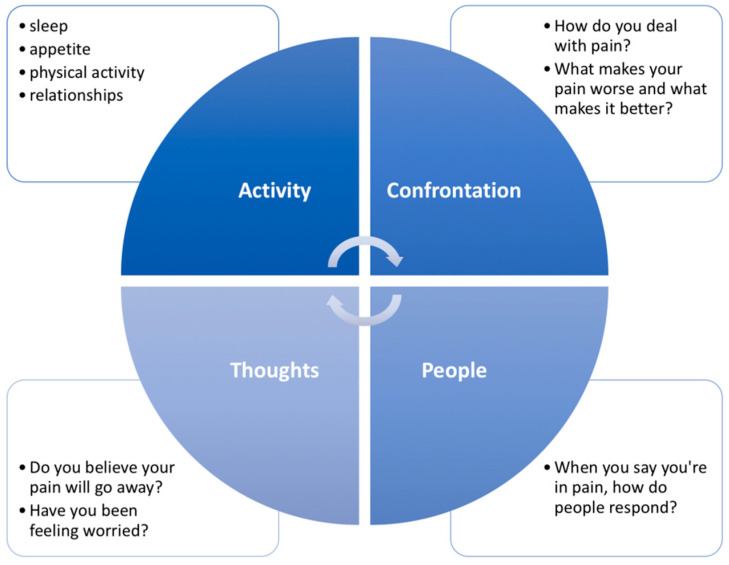
Brief screening model of the psychosocial aspects of the individual with pain based on the ACT-UP (activity, coping, think, upset, people’s responses) model [63].

**Figure 5 diagnostics-13-00062-f005:**
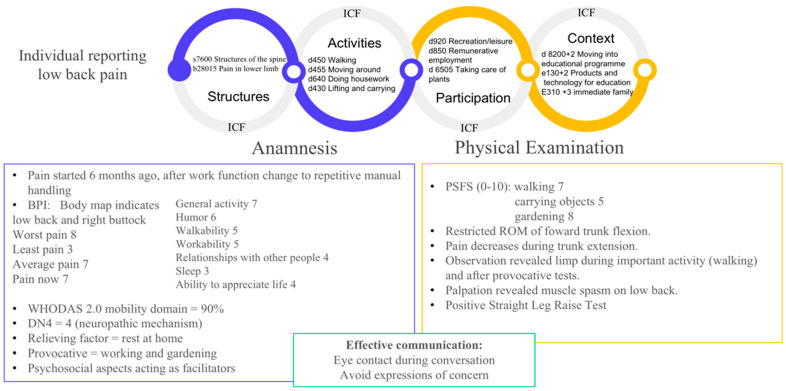
An example of clinical application of the chronic musculoskeletal pain framework.

## Data Availability

Not applicable.

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
