# Peer review of "Clinical Assessment of Chronic Musculoskeletal Pain—A Framework Proposal Based on a Narrative Review of the Literature"

_diagnostics, 2022, doi:10.3390/diagnostics13010062_

Round 1

Reviewer 1 Report

Dear Authors,

the purpose of your study is very interesting, since it is often difficult to manage patients with chronic musculoskeletal pain.

Your manuscript is overall well organized and well written, just few things should be fixed. Some examples are reported below:

Line 149: "insidiously"

Line 227: "for illustration (purposes?)" Eventually rephrase.

Line 243: "the role...has been". Please check all through the manuscript for any subject-verb disagreement.

Line 316: "patient plus presents". What does it mean?

Subsection 5.4 and 5.5 are inverted. Please correct.

Line 362: "employed". Please use a synonym such as "are used"

Line 366-369: please explain better the methods to assess allodynia (such as 4kg of what? or "considering approximately 20ºC").

Line 392: an universal

Line 396-397: please rephrase.

Line 408: "Social factors act as facilitators, since they aid activities and participation". The verb "aid" is not the most appropriate in this context. I suggest you to review the use of some verbs, since some of them are out of context.

Reviewer 2 Report

Reviewer Comments

Thank you very much for the opportunity to review the manuscript submission entitled: “Clinical assessment of chronic musculoskeletal pain - a framework based on current concepts

The objective of this paper is to provide a framework for the clinical assessment of CMP, outlined according to the ICF and based on current pain research. The study is interesting; however, some limitations and constructive comments are pointed out below:

Topic: Appropriate for the scope of the Journal. Manuscript can have more depth to add to the body of knowledge.

Title: include the type of review in the title

Abstract:

·      End the abstract relating your review with clinical significance.

·      Need to include your main search strategy; how many studies were included in this study? What were the criteria?

Introduction

·      Introduction should be emphasised on

o    Present relevance to practice,

o   Present relevant background material

Methods

·      Present key elements of study design early in the paper

·      Discuss literature search strategies, including databases used and MeSH terms

·      Include rationale for inclusion and exclusion criteria

·      Include the level of detail required to replicate the study

Discussion

·      The discussion should present content strengths and limitations in an unbiased manner

·      The discussion should provide a thorough analysis of the literature

·      The discussion should address the implications of the results for evidence-based practice

Round 2

Reviewer 2 Report

The authors have addressed all the comments raised by me. The manuscript has been significantly improved. The study can be accepted in its current form for publication.